# Impact of BNT162b2 Booster Dose on SARS-CoV-2 Anti-Trimeric Spike Antibody Dynamics in a Large Cohort of Italian Health Care Workers

**DOI:** 10.3390/vaccines11020463

**Published:** 2023-02-17

**Authors:** Laura V. Renna, Fabio Bertani, Alessandro Podio, Sara Boveri, Matteo Carrara, Arianna Pinton, Valentina Milani, Giovanni Spuria, Angelica F. Nizza, Sara Basilico, Carola Dubini, Ambra Cerri, Lorenzo Menicanti, Massimiliano M. Corsi-Romanelli, Alexis E. Malavazos, Rosanna Cardani

**Affiliations:** 1Biobank BioCor, IRCCS Policlinico San Donato, San Donato Milanese, 20097 Milan, Italy; 2Residency Program in Clinical Pathology and Clinical Biochemistry, University of Milano, 20157 Milan, Italy; 3Laboratory of Biostatistics and Data Management, Scientific Directorate, IRCCS Policlinico San Donato, San Donato Milanese, 20097 Milan, Italy; 4Endocrinology Unit, Clinical Nutrition and Cardiovascular Prevention Service, IRCCS Policlinico San Donato, San Donato Milanese, 20097 Milan, Italy; 5Scientific Directorate, IRCCS Policlinico San Donato, San Donato Milanese, 20097 Milan, Italy; 6Department of Cardiac Surgery, IRCCS Policlinico San Donato, San Donato Milanese, 20097 Milan, Italy; 7Department of Biomedical Sciences for Health, University of Milano, 20133 Milan, Italy; 8Operative Unit of Laboratory Medicine-Clinical Pathology, Department of Pathology and Laboratory Medicine, IRCCS Policlinico San Donato, San Donato Milanese, 20097 Milan, Italy; 9Department of Biomedical, Surgical and Dental Sciences, University of Milano, 20122 Milan, Italy

**Keywords:** SARS-CoV-2, vaccine, immunity, BNT162b2, COVID-19, antibody, booster dose, anti-Trimeric Spike, natural infection

## Abstract

Accurate studies on the dynamics of Pfizer-Biontech BNT162b2-induced antibodies are crucial to better tailor booster dose administration depending on age, comorbidities, and previous natural infection with SARS-CoV-2. To date, little is known about the durability and kinetics of antibody titers months after receiving a booster dose. In this work, we studied the dynamic of anti-Trimeric Spike (anti-TrimericS) IgG titer in the healthcare worker population of a large academic hospital in Northern Italy, in those who had received two vaccine doses plus a booster dose. Blood samples were collected on the day of dose 1, dose 2, then 1 month, 3 months, and 6 months after dose 2, the day of the administration of the booster dose, then 1 month and 3 months after the booster dose. The vaccination immunogenicity was evaluated by dosing anti-TrimericS IgG titer, which was further studied in relation to SARS-CoV-2 infection status, age, and sex. Our results suggest that after the booster dose, the anti-TrimericS IgG production was higher in the subjects that were infected only after the completion of the vaccination cycle, compared to those that were infected both before and after the vaccination campaign. Moreover, the booster dose administration exerts a leveling effect, mitigating the differences in the immunogenicity dependent on sex and age.

## 1. Introduction

SARS-CoV-2 (severe acute respiratory syndrome-coronavirus-2) is a beta-coronavirus that has been defined as the causal agent of COVID-19 (Coronavirus Disease 2019), a spreading pneumonia that originated from the Hubei province of China at the end of 2019 [1]. In March 2020, the World Health Organization declared COVID-19 as a global pandemic and, as of 31 December 2022, reported more than 655 million cases and more than 6.6 million deaths all over the world.

The symptoms may vary, ranging from asymptomatic infections to mild symptoms (fever/chills, cough, shortness of breath, fatigue, loss of taste/smell, headaches, runny nose, sore throat, nausea/vomiting, and diarrhea), to more severe cases of bilateral pneumonia, acute respiratory failure, and death.

Due to the international public health emergency following the SARS-CoV-2 outbreak in early January 2020, unprecedented efforts were made to quickly develop a vaccine that could be safe and effective in preventing the spread of contagion.

Originally approved through temporary emergency use authorization in December 2020, BNT162b2 (Comirnaty, BioNTech and Pfizer) is a nucleoside-modified RNA (modRNA) that encodes the entire SARS-CoV-2 Spike protein (S) [2,3,4,5]. The nucleic acid is enclosed in a lipid nanoparticle and is modified by two proline mutations to lock the protein in its prefusion conformation. BNT162b2 is administered intramuscularly in a two-dose regimen. The efficacy and safety of Comirnaty has been extensively reported on and has led to a significant reduction in severe cases and deaths [6,7].

In Italy, as soon as Comirnaty was approved for administration by an emergency law act, a vaccination campaign started at the end of 2020. In early 2021, vaccination was made mandatory for healthcare workers and the residents of long-term care facilities [8]. Subsequently, the vaccination was extended to the remaining population. Extended trial follow-up and real-world effectiveness studies documented the vaccination-induced antibody kinetics, often resulting in contrasting results [9,10,11]. Some groups have found rapidly waning immunity over time, while others have demonstrated a stable, persistent IgG titer [12,13,14]; thus, the nature, durability, and stability of the neutralizing antibodies produced after vaccination remains not fully understood. In October 2021, a third (booster) dose was approved for administration [15,16].

Accurate studies on the dynamics of vaccination-induced antibodies are crucial to better tailor future vaccine doses and the time of booster administration, depending on age, comorbidities, and previous natural infection with SARS-CoV-2 [17,18,19].

To date, although several follow-up studies on antibody dynamics have been published, little is known about the durability and kinetics of the antibody titers after receiving booster doses [18,20,21,22].

Since the Ministry of Health in Italy declared that over one quarter of the population has been infected starting from January 2020, the level of antibodies elicited by natural SARS-CoV-2 infection must be taken into account in order to prevent misleading results in antibodies kinetics studies following vaccination.

In this work, we studied the dynamic of anti-Trimeric Spike (anti-TrimericS) IgG titer in a large cohort of healthcare workers from an Academic Hospital in Northern Italy during the first vaccination cycle (dose 1 and dose 2), plus the first booster dose. IgG titers have been studied in relation to SARS-CoV-2 infection, as well as to population characteristics.

## 2. Materials and Methods

### 2.1. Study Design

The population included in this work was enrolled in the VARCO-19 study, a prospective observational study that started in January 2021 and ended in March 2022 at the Istituto di Ricovero e Cura a Carattere Scientifico (IRCCS), Policlinico San Donato [23]. The protocol was approved by the Ethics Committee of the IRCCS Lazzaro Spallanzani (protocol code 48/2021/spall/PU/403–2021) and the participants provided written informed consent.

Participants were enrolled between January 2021 and February 2021 among the personnel of the IRCCS Policlinico San Donato who participated in the vaccine campaign, receiving the BNT162b2 mRNA vaccine. Among the 1232 subjects that were initially enrolled in the VARCO-19 study, we analyzed the antibody response of those who completed the project (*n* = 698). The participants belonged to 6 different professional categories: medical doctors, non-medical doctors (i.e., biologists, physicists, and engineers), nurses and social health workers, auxiliary health workers, administrative staff members, and non-health care workers (i.e., university students, barmen, hospital volunteers). Blood sample collections were performed at the following time points: the day of the administration of the BNT162b2 dose 1 (D1), the day of the administration of dose 2 (D2, 3–4 weeks after D1), then 1 month (D2M1), 3 months (D2M3), and 6 months (D2M6) after dose 2. Moreover, the samples were also collected on the day of the administration of the booster dose (BD), then 1 month (BDM1) and 3 months (BDM3) after the booster dose (Figure 1a). At each time point, the data on positive SARS-CoV-2 nasopharyngeal swabs were obtained from participants.

In order to evaluate if the SARS-CoV-2 infection occurred before and/or throughout the study, the subjects were screened with qualitative assays for anti-nucleocapsid IgG (anti-N IgG) at the beginning and at the end of the study (Figure 1a). The subjects who resulted positive anti-N IgG at both time points were further tested on intermediate samples (D26M or BD). These results were matched with the information given by the participants about their infection status (i.e., if they had resulted positive with an antigenic and/or molecular swab test after D1 during the observation period). On the basis of the results obtained, the participants were divided in four SARS-CoV-2 infection groups (Figure 1b): (i) never-infected subjects (NN, Negative/Negative), if the anti-N IgG analysis showed negative results both before and after the vaccination campaign and no positive swabs tests were reported; (ii) individuals infected during the vaccination campaign (NP, Negative/Positive), if the anti-N IgG analyses showed a negative result before the vaccination campaign but a positive result was obtained at the end of the study and/or at least one positive swab test was reported; (iii) individuals infected before and after the vaccination campaign (PP, Positive/Positive), if the anti-N IgG analyses showed a positive result both before and at the end of the observation period, with a negative result at an intermediate timepoint and/or at least one positive swab test was reported; and (iv) subjects infected before the beginning of the vaccination campaign (PN, Positive/Negative), if the anti-N IgG analyses showed a positive result before but not at the end of the vaccination campaign, and no positive swab tests were reported. Subjects positive for all three anti-N IgG tests, without a documented infection after D1, were assumed to have a long-lasting anti-N titer and were subsequently assigned to the PN group.

### 2.2. Specimens Collection and Biobanking

Blood samples were collected by phlebotomists using a winged drawing system, equipped with a 21-gauge needle (BD Vacutainer) connected to a needle holder (BD Vacutainer). The biological samples were then immediately transferred to the BioCor Biobank of the IRCCS Policlinico San Donato to be processed and stored by qualified staff members. Standard operating procedures (SOP) were prepared to ensure uniform blood sample collection, manipulation, storage, and retrieval. For serum preparation, the blood was collected in clot activator tubes (CAT, BD Vacutainer) and left for at least 30 min to completely coagulate, according to the manufacturer’s instructions. The tubes were then centrifuged at 1500 RCF at room temperature and the sera were separated manually from the blood clot and aliquoted in cryovials with a one-dimensional barcode. The aliquots were then transferred at −80 °C until immunological analysis. For the registration, control, and movement of the biological material, the specialized software EasyTrack2 with barcode-scanning technology was used.

### 2.3. Quantitative Analysis of Anti-Trimeric Spike IgG antibodies

The serum samples were analyzed in the Laboratory of Medicine and Clinical Pathology of the IRCCS Policlinico San Donato. The anti-Trimeric Spike protein IgG titer was determined on serum samples by using LIAISON^®^ SARS-CoV-2 TrimericS IgG chemiluminescence assay CLIA (DiaSorin, Italy), performed on a LIAISON^®^ XL platform (DiaSorin, Italy). The antigen used for the IgG determination is a recombinant Trimeric Spike Glycoprotein, which is the native form of the SARS-CoV-2 Spike protein, allowing for the detection of a wider range of antibodies directed against different epitopes [24]. The test has 98.7% clinical sensitivity 15 days post-positive PCR and a 99.5% clinical specificity. As reported by the manufacturer’s instructions, the assay range is 4.81–2080 BAU/mL (binding antibody units/mL, as stated by WHO 20/136 International Standard). A positive result (≥33.8 BAU/mL) indicates the presence of antibodies following vaccination and/or infection. When an antibody determination was above the upper limit of the assay range, a 1:20 dilution with specific LIAISON^®^ TrimericS IgG Diluent Accessory buffer (DiaSorin, Italy) was used. Due to the limit of the linear range at the 1:20 dilution, sporadic titers of >40,000 BAU/mL were obtained. Since the occurrence of that event was uncommon, no further dilutions were performed, and a titer of BAU/mL was assigned to those specimens.

### 2.4. Qualitative Assessment of Anti-Nucleocapsid IgG Antibodies

A qualitative assessment of anti-N IgG was performed at the beginning of the observation period (D1 or D2) in order to screen if any SARS-CoV-2 infections had occurred before vaccination (Figure 1). The serum samples were analyzed in the Laboratory of Medicine and Clinical Pathology of the IRCCS Policlinico San Donato. The analysis was performed with the Elecsys Anti-SARS-CoV-2 kit (Roche, Italy), equipped on the automated system Cobas e 601 (Roche, Italy). The test is based on the electrochemiluminescence immunoassay ECLIA principle. The results are calculated according to a cut-off index (COI) obtained by two-point calibration. Whether the COI is equal or greater than 1.0, the analyzed serum is reactive; hence, it is considered positive.

At the end of and during the observation period, the anti-N IgG titer was determined again to evaluate whether a participant had been infected with SARS-CoV-2 (Figure 1). These analyses were performed in the Laboratory of the BioCor Biobank, since Cobas e 601 in the Medical Laboratory was replaced. An ELISA test kit was used according to the manufacturer’s instructions (ab277285 SARS-CoV-2 (COVID19) IgG ELISA Kit, Abcam). The concordance between the Roche and Abcam tests was evaluated according to the Clinical & Laboratory Standards Institute’s (CLSI) EP12-A2 guidelines. The samples were diluted 1:100 with an appropriate kit diluent and were then plated onto ELISA microwells. For each analysis, positive, negative, and cut-off controls supplied by the manufacturer were used. The results were measured by reading the optical density at 450 nm and arbitrary units (AU) were assigned to each sample according to the kit equation, considering the cut-offs optical density. AU values lower than 9 were considered negative, values between 9 and 11 were considered uncertain, and those higher than 11 were determined to be positive. The uncertain samples were excluded from the study.

### 2.5. Statistical Analysis

Categoric variables are presented as frequency and percentages while continuous variables are shown as means ± standard deviation (SD) and as a median and interquartile range. The antibody levels are expressed as a geometric mean ± standard error (SE). The comparison of the continuous values between the groups was done with a nonparametric Kruskal–Wallis test. A univariate linear regression model was used to evaluate the association between the absolute variation between 1 and 3 months of the IgG-Trimerics antibodies after the second dose (or after the booster dose), and sex, age, and the IgG antibody level, starting from 1 month after dose 2 (or after the booster dose). A multivariable linear regression model to account for possible confounding factors (age, sex, and the IgG antibody level 1 month after the vaccinations: dose 2 or the booster dose) was used to evaluate the 3-month difference in the absolute variation of the titer levels and infection status. Least square means (±SE) were reported. The intragroup difference of variation after the second and after the booster dose was analyzed by a longitudinal mixed regression model. The null hypothesis will be refused with *p* < 0.05. All statistical analyses were done with SAS version 9.4 (SAS Institute, Cary, North Carolina).

## 3. Results

### 3.1. Population Characteristics

The study population characteristics are shown in Table 1. Participants (*n* = 698) had a median age of 43.27 ± 11.80 and 66.8% were female (*n* = 466).

At the beginning of the study, 23.1% (*n* = 161/698) of the subjects showed a positive serological IgG response against N-Protein, while 29.2% (*n* = 204/698) resulted positive anti-N IgG at the end of the study.

According to the serological anti-N IgG results and to data on SARS-CoV-2 nasopharyngeal swabs, subjects were classified as no SARS-CoV-2 infection (NN) (*n* = 400), SARS-CoV-2 infection before vaccination (PN) (*n* = 94), SARS-CoV-2 infection after vaccination (NP) (*n* = 137), and SARS-CoV-2 infection before and after vaccination (PP) (*n* = 67). No statistical difference was found in these groups stratified by population characteristics, as shown in Table 1.

### 3.2. Anti-Trimeric Spike IgG Titer Dynamic throughout Vaccination Campaign

#### 3.2.1. Overall Dynamic of Anti-Trimeric Spike IgG

The anti-Trimeric Spike IgG titer was assessed at all the time points (Figure 2 and Figure 1a). As expected, when considering the overall IgG dynamics, the geometric mean titer increased progressively from D1 until D2M1, reaching its peak value at 2766.60 ± 107.27 BAU/mL, one month after the conclusion of the first vaccination cycle. Then, the IgG value progressively waned until the BD, where the overall nadir reached 265.28 ± 11.98 BAU/mL. This decline was more pronounced between D2M1 and D2M3. The highest IgG value was reached at BDM1, with a second peak value of 6222.81 ± 227.7 BAU/mL that was twice the value of the previous one. As expected, it was followed by further decrease in the anti-Trimeric Spike IgG.

#### 3.2.2. Anti-Trimeric Spike IgG Titer Dynamic According to SARS-CoV-2 Infection Status

When considering the anti-Trimeric Spike IgG titer before the booster dose administration, as expected, individuals with hybrid immunity due to the combination of a SARS-CoV-2 infection and vaccination (PN and PP) had a statistically significant higher BAU/mL titer compared to other groups (Figure 3a,b). In detail, post hoc tests showed a statistically significant difference at D1, D2, D2M1, D2M3, D2M6 and the BD, comparing NN vs. PP (*p* < 0.0001), PN vs. NP (*p* < 0.0001), NP vs. PP (*p* < 0.0001), and PN vs. NN (*p* < 0.0001) (Figure 3). The significant differences between the groups at BDM1 were for PN vs. NN (*p* = 0.01) and PN vs. NP (*p* = 0.002), and finally at BDM3 for PN vs. NP, NN vs. NP, PN vs. PP (*p* < 0.0001), and PN vs. PP (*p* = 0.004) (Figure 3a). Interestingly, in the PN and PP subjects, the peak value of the antibody response was reached before the administration of dose 2 (D2), while in the remaining groups was reached at D2M1. Moreover, when comparing the IgG peak values, the PN individuals exhibited significantly higher IgG levels than those observed in the other groups (PN and PP peaks were at D2: 6975.70 ± 924.49 and 8891.60 ± 1345.76, and the NN and NP peaks were at D2M1: 2079.06 ± 90.50 and 2318.88 ± 167.62). When considering the IgG dynamics after the booster dose administration, the NP individuals showed a higher BAU/mL titer compared to all other groups (BDM1: PN vs. NP *p* = 0.002; BDM3: NN vs. NP *p* < 0.0001, PN vs. NP *p* < 0.0001, NP vs. PP *p* < 0.0001, and PN vs. PP *p* = 0.004). In these subjects, the antibodies progressively increased from the booster dose until the end of the observation period, reaching the higher levels observed in the study (11,285.00 ± 882.86 BAU/mL; fold 1.47 CI 95% [1.15 ÷ 1.88]). On the contrary, for PN and NN, the IgG peak values were observed at BDM1 and, notably, the NN individuals showed a higher BAU/mL titer compared to that observed in the individuals infected before the vaccination campaign (PN and PP) (NN vs. PN *p* = 0.001). The BDM1 peak value was followed by a statistically significant decrease both in PN and NN individuals (PN fold 0.54 CI 95% [0.42 ÷ 0.71]; NN fold 0.54 CI 95% [0.46 ÷ 0.61]). Surprisingly, the re-infected individuals (PP) exhibited an IgG peak value at BDM1, similar to those observed in the individuals that were never infected or infected only before the vaccination campaign. However, the PP subjects did not show a statistically significant decrease in the antibody titer at BDM3 (fold 0.86 CI 95% [0.62 ÷ 1.20]) and exhibited a higher BAU/mL titer than those infected only before the observation period (PN vs. PP *p* < 0.0001) (Figure 3a).

#### 3.2.3. Anti-Trimeric Spike IgG Titer Dynamic According to Age and Sex

Figure 4 illustrates the antibody titer dynamic in relation to age. When considering the time points before the booster dose, the subjects younger than 40 years of age showed higher BAU/mL values compared to the older subjects (respectively, the overall comparison showed D2 *p* < 0.0001, D2M1 *p* = 0.0004, D2M3 *p* = 0.0001, and D2M6 *p* < 0.0001). In detail, a post hoc analysis showed a significant difference at D2M1 between <30 vs. >60 (*p* = 0.007), 30–40 vs. 50–60 (*p* = 0.047) and <30 vs. 50–60 (*p* = 0.002).

Interestingly, this higher antibody response in the younger subjects was no longer observable after the booster dose. Indeed, at BDM1, the age groups <30, 30–40 and 40–50 showed mean titers lower than those observed in the age groups 50–60 and >60 (post hoc analysis: 40–50 vs. 50–60 *p* = 0.01 and 30–40 vs. 50–60 *p* = 0.03). At the end of the observation period (BDM3), no statistically significant differences in the BAU/mL were observed between the different age groups.

Despite being different in the quantitative IgG response, all of the age groups showed similar IgG titer dynamics, with an antibody peak value at D2M1 and a second peak value at BDM1 (Figure 4b).

When considering the subjects grouped by sex, females (66.8%) registered statistically significant higher BAU/mL values compared to males at the time points before the booster dose (D2: 780.21 ± 58.52 BAU/mL vs. 581.81 ± 69.04 BAU/mL, *p* = 0.005; D2M1: 2962.82 ± 138.18 BAU/mL vs. 2395.01 ± 163.57 BAU/mL *p* < 0.001; D2M3: 981.39 ± 44.43 BAU/mL vs. 894.86 ± 62.51 BAU/mL, *p* = 0.01; D2M6: 513.87 ± 442.60 BAU/mL vs. 442.60 ± 34.11 BAU/mL, *p* = 0.02; and BD: 280.50 ± 14.71 BAU/mL vs. 236.84 ± 20.42 BAU/mL *p* = 0.004) (Figure 5a). After the administration of the booster dose, no statistically significant difference was observed between females and males. The antibody response dynamic was similar in both groups, with superimposable antibody peak values at D2M1 and around BDM1 (Figure 5b).

### 3.3. Effects of Variables in Antibody Titer Waning after Dose 2 and after Booster Dose

A univariate linear regression evaluating the 3-month difference in the absolute variation of the IgG-TrimericS antibody levels, starting 1 month after dose 2, demonstrated no correlations with sex and age (Table 2). However, as expected, a statistically significant correlation between the antibody titer waning, the IgG levels at 1 month after dose 2, and the infection status groups was observed (*p* < 0.0001). The multivariable linear regression, adjusted to sex, age and the IgG antibody level at D2M1, was conducted using the NN group as a reference and showed a significant difference in the subjects infected before the observation period and in those reinfected during the study (PN: LSmeans ± SE −1571.83 ± 346.06, *p* < 0.0001 and PP: LSmeans ± SE −1975.32 ± 413.20 *p* = 0.003). Indeed, the NP subjects exhibited a similar waning with respect to the reference group (LSmeans ± SE: −3212.88 ± 272.49, *p* = 0.76).

When considering the antibody waning after the booster dose, the univariate linear regression showed similar results to those obtained for dose 2 (Table 2). However, the multivariable linear regression conducted using the never infected group (NN) as a reference showed a statistically significant difference only in the subjects infected during the observation period (NP: LSmeans ± SE 5213.61 ± 650.72, *p* < 0.0001 and PP: LSmeans ± SE −1262.72 ± 882.91, *p* = 0.004). On the contrary, the subjects infected only before the vaccination campaign had a similar wane to the never infected group (LSmeans ± SE: −3987.68 ± 716.81 *p* = 0.99).

Table 3 illustrates the fold changes between 1 month and 3 months after dose 2 and after the booster dose. The intragroup comparison demonstrated a significant difference when comparing the calculated fold changes after dose 2 and the booster dose in three of the four infection status groups (NN, PP, PN: *p* < 0.0001). The borderline statistical significance in the subjects infected exclusively during the observation period (*p* = 0.057) is allegedly due to the high variability in the antibody titer after the booster dose administration.

## 4. Discussion

In this work, we have analyzed the immune response to three doses of the Pfizer-Biontech BNT162b2 vaccination in a large cohort of healthcare workers from an Academic Hospital in Northern Italy. The vaccination immunogenicity was evaluated by dosing anti-TrimericS IgG titer, which was further studied in relation to SARS-CoV-2 infection status, age, and sex. Anti-TrimericS IgG levels are known to correlate with the neutralization titer, thus reflecting in vivo protection against SARS-CoV-2 infection and/or symptomatic disease [24,25,26,27,28,29].

Infection status was assessed serologically at the beginning and at the end of the observation period, determining the IgG seropositivity to the SARS-CoV-2 anti-N protein, and on the basis of the reported positive swabs. Indeed, it is known that anti-N IgG positivity correlates with natural exposure to SARS-CoV-2; hence, it is considered a good indicator of previous infections [30,31]. Recent findings have demonstrated that the seroreversion of anti-N antibodies occurs at a minimum of 7–8 months after seroconversion, which typically takes place 14–21 days after SARS-CoV-2 infection [32,33,34,35,36,37]. A serological assessment of previous SARS-CoV-2 infections in vaccinated individuals is crucial to identify pauci- and asymptomatic subjects that never received a diagnosis via a molecular or antigenic swab test.

According to our data, the re-infection rate registered at the end of our study was 9.6% (67/698). This is similar to those reported by other authors, in comparable cohorts, after the B.1.1.529 Omicron variant outbreak [38,39,40,41]. Indeed, based on both the anti-N and anti-TrimericS IgG titers, and on positive swab reports, we determined that most of the infections occurring during the observation period were registered in the winter of 2021/2022 (December 2021–February 2022). That period was characterized by an increased viral circulation and novel, more contagious Omicron variants rising [42]. Notably, infection status was not associated with population characteristics such as gender, age, and occupation.

As previously reported by other authors, the overall anti-TrimericS IgG titer in our cohort demonstrated a peak value 1 month after dose 2 administration, followed by a progressive decline until the administration of the booster dose, 9 months after dose 1 [43]. The booster dose yielded a great production of antibodies, resulting in a higher peak value observed 1 month after its administration. Once again, 3 months after the booster dose, antibody levels were found to be declining, even though they remained higher than the level found throughout the observation period after the first vaccination cycle completion. These results are in line with what was observed by Faas and colleagues (2022), who analyzed anti-RBD IgG titer for 7 months after the booster dose. Indeed, authors observed a significant decrease in the antibody response 4 months after the booster dose administration, that subsequently remained steady afterward [44].

Interesting data have been obtained when considering the SARS-CoV-2 infection status. As expected, the anti-TrimericS IgG level before the booster dose in vaccinated individuals with history of SARS-CoV-2 infection was higher than that of subjects without infection. Indeed, the individuals with hybrid immunity demonstrated a stronger immune response, reaching faster and higher antibody levels compared to the subjects that were not infected at that time. Individuals without a history of SARS-CoV-2 infection before the beginning of the study (NN and NP) showed a stronger response 1 month after dose 2 compared to the response they had after dose 1. Moreover, in these subjects, the IgG peak value was significantly lower compared to the previously infected individuals (PN and PP). In vaccinated participants with a history of SARS-CoV-2 infection, anti-TrimericS IgG levels remained higher during the 8 months after the completion of the first vaccination cycle, although demonstrating similar waning kinetics to that observed in non-infected subjects. These results are in line with what was observed in other studies [44,45].

As expected, after the booster dose, the anti-TrimericS IgG response was stronger than those elicited by the previous doses, indicating that vaccination elicits robust SARS-CoV-2-specific immune memory, regardless of prior infection. This finding agrees with what has been observed by other authors [42,46,47]. Importantly, the subjects that were infected during the observation period were characterized by an evident increase in anti-TrimericS IgG levels until the end of the study (3 months after the booster dose). This trend is due to the hybrid immunity which was present in these individuals after the booster dose. However, in the re-infected individuals (PP), the antibody level did not continue to rise, but a slight decrease was observable 1 month after the booster dose, even if this decrease was less evident than in non-infected individuals and in those infected before the vaccination cycle. Indeed, by determining the fold decrease in antibody titer 1 month to 3 months after the booster dose, no statistically significant decrease was observed in the re-infected individuals, suggesting that the IgG titer remained roughly stable during this period. This is again justified by the high prevalence of the SARS-CoV-2 infections that occurred in January/February 2022, determining both the increasing or stabilizing antibody levels [42].

Data obtained by analyzing the anti-TrimericS IgG dynamics according to infection status suggest that natural infection was acquired only after the vaccination resulted in a higher anti-TrimericS IgG production, compared to what occurs in the subjects that were infected both before and during the vaccination campaign. One reason that possibly explains this fact is that never-infected individuals usually develop symptomatic COVID-19, and this may result in a stronger immune response, positively affecting the antibody production [48].

Interestingly, we found that after the administration of the booster dose, the mean BAU/mL in never-infected individuals and subjects that were infected before the vaccination campaign was not statistically different. This demonstrates the waning immunity conferred by natural infection over time, indicating the importance of vaccination in eliciting a strong immune response that reaches the same antibody titer observed in naturally infected individuals, confirming what was previously observed [44,49].

The data obtained from our cohort provided that females had a significantly higher antibody response compared to males 1 month after the second dose, after which similar IgG titers were observed. Likewise, younger subjects (aged <40) demonstrated higher levels of anti-TrimericS antibodies until the booster dose compared to older individuals, as profusely demonstrated by other authors stating that the vaccination immunogenicity was higher in women and younger subjects [50,51]. Interestingly, a surprising inversion was observed one month after the booster dose administration, when the 30–40 and 40–50 groups showed a lower BAU/mL compared to the 50–60 group, while at the end of the observation period, no statistically significant differences in BAU/mL in different age groups were observed. Notably, these differences were not justified by the different prevalence of infection in age or gender groups. These results demonstrate that the booster dose administration exerts a leveling effect, mitigating the differences in immunogenicity related to sex and age. Moreover, the higher levels of the anti-TrimericS IgG observed in the older population suggest an important role of booster doses in maintaining the immunity of elders, as previously reported by other authors [46,52,53].

Analyzing the antibody waning, our results indicate that a 1–3 months decrease in the antibody titer after dose 2 and after the booster was not correlated to sex and age, but exclusively to the SARS-CoV-2 infection status. Previously infected subjects demonstrated a slower IgG titer decrease after dose 2, compared to infection-naïve subjects, while, on the contrary, those infected during the observation period exhibited a slower decrease in antibody titer after the booster dose. Interestingly, the subjects infected before the vaccine dose 1 administration had a waning kinetic after the booster dose, comparable to what was observed in the never-infected individuals and confirming the importance of booster dose administrations, even in subjects naturally infected by SARS-CoV-2 [44]. More importantly, when comparing the antibody titer decrease within the infection status groups, never-infected subjects showed a slower waning kinetic after the booster dose administration, with respect to what was observed after dose 2. Again, this result underlines the importance of booster doses in maintaining the persistently high anti-TrimericS IgG levels in the general population [44,45,54,55].

There are some limitations to acknowledge in this study. First, our cohort of participants had a median age lower than the general population. Individuals older than 60 years, who are more likely to develop severe COVID-19, are under-represented. Second, to assess vaccine-induced immunity, only the anti-TrimericS IgG antibodies were determined. It should be taken into consideration that these antibodies are only a part of the immune response induced by vaccination and/or infection, therefore, other components should be addressed to completely evaluate the vaccination efficacy (i.e., neutralizing antibodies and the mucosal IgA and cellular response). Finally, we had no data regarding the SARS-CoV-2 variants responsible for the infections during the observation period, thus, we were not able to correlate the antibody titer to the different SARS-CoV-2 viral strains.

## 5. Conclusions

In conclusion, the results of this study show that the booster dose elicited a stronger immune response than the first two doses of BNT162b2, mitigating the differences previously observed both between males and females, and between age groups. The booster dose slowed down the waning of antibody titer, independently of infection status, confirming its importance in preserving a high antibody titer over time.

## Figures and Tables

**Figure 1 vaccines-11-00463-f001:**
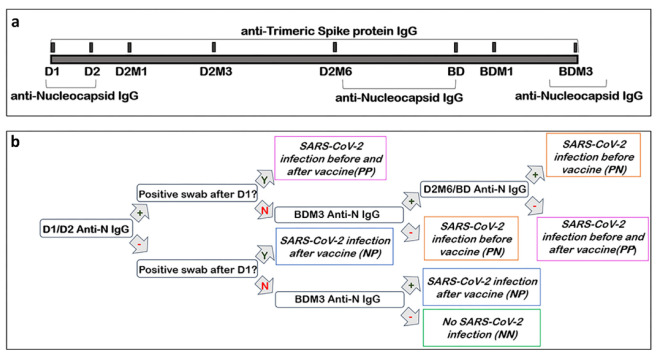
Illustration of study design: (**a**) timeline and methods; and (**b**) schematic representation of the methods used to determine the SARS-CoV-2 infection group allocation. D1, dose 1; D2, dose 2; D2M1, 1 month after dose 2; D2M3, 3 months after dose 2; D2M6, 6 months after dose 2; BD, booster dose; BDM1, 1 month after booster dose; BDM3, 3 months after booster dose; SARS-CoV-2, severe acute respiratory syndrome-coronavirus-2; anti-N IgG, anti-nucleocapsid IgG; NN, never-infected subjects; PN, subjects infected before the beginning of the vaccination cycle; NP, individuals infected during the vaccination campaign; and PP, individuals infected before and after the vaccination campaign.

**Figure 2 vaccines-11-00463-f002:**
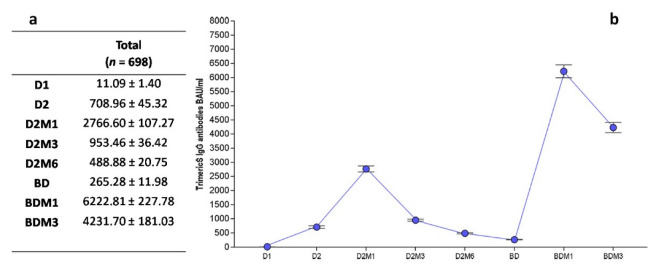
Anti-Trimeric Spike IgG overall mean titer (**a**), and dynamic (**b**) (*n* = 698). D1, dose 1; D2, dose 2; D2M1, 1 month after dose 2; D2M3, 3 months after dose 2; D2M6, 6 months after dose 2; BD, booster dose; BDM1, 1 month after booster dose; and BDM3, 3 months after booster dose.

**Figure 3 vaccines-11-00463-f003:**
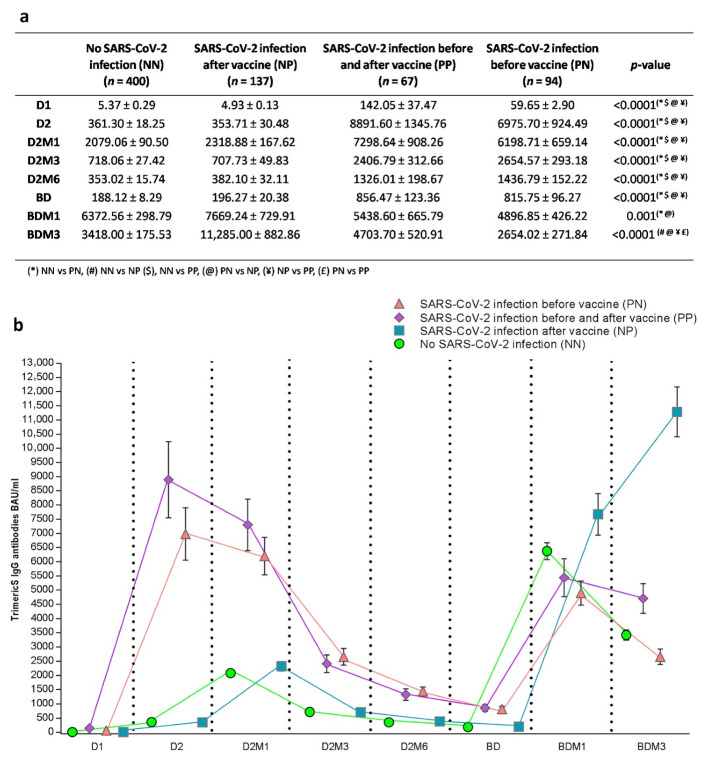
Anti-Trimeric Spike IgG geometric mean titer (**a**), and dynamic (**b**) before and after booster vaccination according to SARS-CoV-2 infection status (mean ± SE). The nonparametric Kruskal–Wallis test was used for comparison of continuous values between groups. D1, dose 1; D2, dose 2; D2M1, 1 month after dose 2; D2M3, 3 months after dose 2; D2M6, 6 months after dose 2; BD, booster dose; BDM1, 1 month after booster dose; BDM3, 3 months after booster dose; and SARS-CoV-2, severe acute respiratory syndrome-coronavirus-2.

**Figure 4 vaccines-11-00463-f004:**
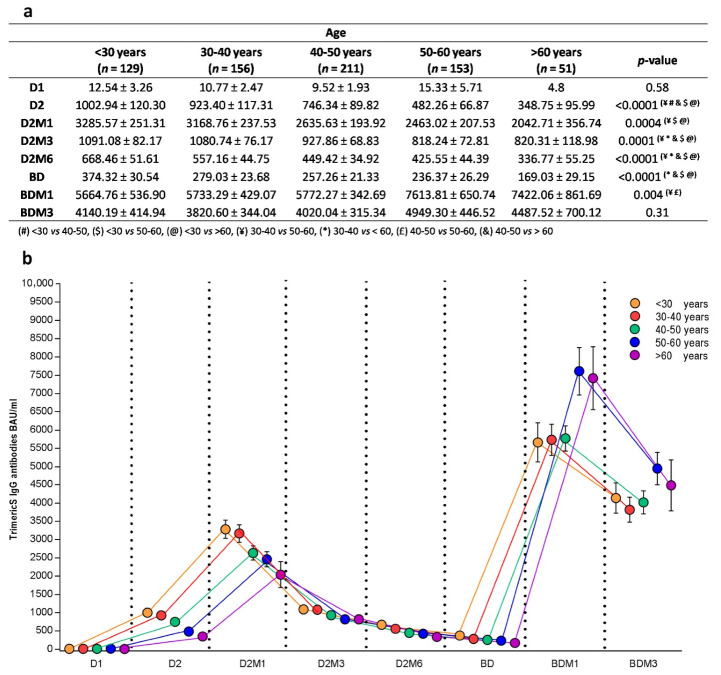
Anti-Trimeric Spike IgG geometric mean titer (**a**), and dynamic (**b**) before and after booster vaccination according to age group (mean ± SE). The nonparametric Kruskal–Wallis test was used for comparison of continuous values between groups. D1, dose 1; D2, dose 2; D2M1, 1 month after dose 2; D2M3, 3 months after dose 2; D2M6, 6 months after dose 2; BD, booster dose; BDM1, 1 month after booster dose; BDM3, 3 months after booster dose; and SARS-CoV-2, severe acute respiratory syndrome-coronavirus-2.

**Figure 5 vaccines-11-00463-f005:**
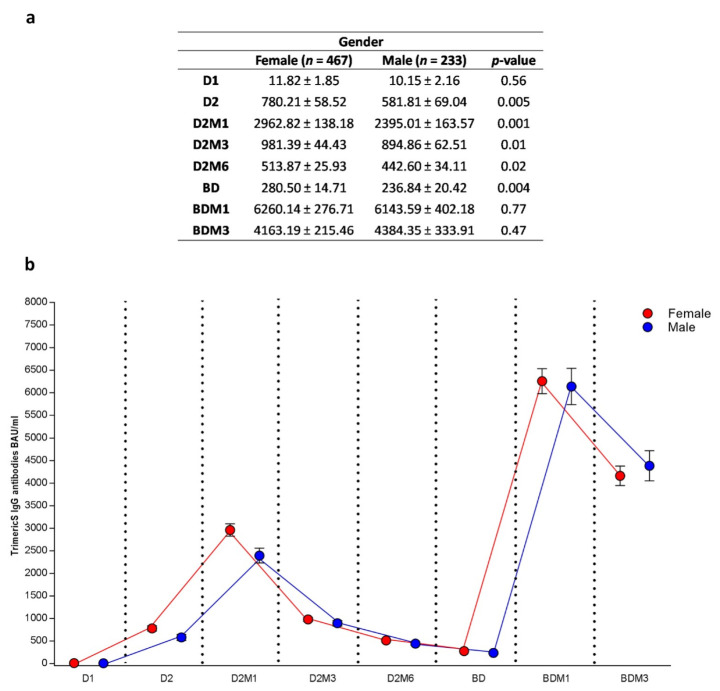
Anti-Trimeric Spike IgG geometric mean titer (**a**) and dynamic (**b**) before and after booster vaccination according to gender (mean ± SE). The nonparametric Kruskal–Wallis test was used for comparison of continuous values between groups. D1, dose 1; D2, dose 2; D2M1, 1 month after dose 2; D2M3, 3 months after dose 2; D2M6, 6 months after dose 2; BD, booster dose; BDM1, 1 month after booster dose; BDM3, 3 months after booster dose; and SARS-CoV-2, severe acute respiratory syndrome-coronavirus-2.

**Table 1 vaccines-11-00463-t001:** Population characteristics.

	No SARS-CoV-2 Infection (NN) (*n* = 400)	SARS-CoV-2 Infection after vaccine (NP) (*n* = 137)	SARS-CoV-2 Infection before and after Vaccine (PP) (*n* = 67)	SARS-CoV-2 Infection before Vaccine (PN) (*n* = 94)	Total (*n* = 698)	*p*-Value
**Age. Median (percentile)**	44.04 ± 12.18; 45.00 (35.00–53.00)	41.79 ± 11.61; 42.00 (33.00–50.00)	41.87 ± 9.56; 42.00 (35.00–50.00)	43.16 ± 11.77; 43.00 (32.00–52.00)	43.27 ± 11.80; 44.00 (34.00–52.00)	0.20
**Female. *n* (%)**	276 (69.0)	82 (59.9)	45 (67.2)	63 (67.0)	466 (66.8)	0.27
**Age *n* (%)**						
(19–30)	69 (17.3)	31 (22.6)	10 (14.9)	18 (19.2)	128 (18.3)	0.38
(30–40)	84 (21.0)	32 (23.4)	19 (28.4)	21 (22.3)	156 (22.4)	
(40–50)	114 (28.5)	43 (31.4)	24 (35.8)	29 (30.9)	210 (30.1)	
(50–60)	99 (24.8)	23 (16.8)	13 (19.4)	18 (19.2)	153 (21.9)	
(>60)	34 (8.5)	8 (5.8)	1 (1.5)	8 (8.5)	51 (7.3)	
**Occupation Status**						
Administrative Staff Members	76 (19.1)	20 (14.6)	5 (7.5)	13 (13.8)	114 (16.4)	0.09
Auxiliary Health Workers	20 (5.0)	9 (6.6)	4 (6.0)	4 (4.3)	37 (5.3)	
Medical Doctors	78 (19.6)	38 (27.7)	11 (16.4)	17 (18.1)	144 (20.7)	
Non Medical Doctors	74 (18.6)	26 (19.0)	10 (14.9)	13 (13.8)	123 (17.7)	
Nurses and Social Health Workers	126 (31.6)	39 (28.5)	33 (49.3)	39 (41.5)	237 (34.0)	
Non-Health Care Workers	25 (6.3)	5 (3.7)	4 (6.0)	8 (8.5)	42 (6.0)	

**Table 2 vaccines-11-00463-t002:** Univariate and multivariable linear regression to evaluate the 3-month difference in absolute variation of IgG-Trimeric Spike antibody levels starting from 1 month after dose 2 (D2M1) or 1 month after booster dose (BDM1).

	After Second Dose		After Booster Dose	
	Univariate	Multivariable	Univariate	Multivariable
	*p*-Value	*p*-Value	*p*-Value	*p*-Value
**Sex**	0.07	0.084	0.705	0.613
**Age**	0.73	0.632	0.476	0.174
**IgG antibody level 1 month after dose**	<0.0001	<0.0001	<0.0001	<0.0001
**SARS CoV-2 infection groups**	<0.0001	<0.0001	<0.0001	<0.0001
**SARS-CoV-2 NP**		0.76		<0.0001
**SARS-CoV-2 PP**		0.003		0.004
**SARS-CoV-2 PN**		<0.0001		0.99
**SARS-CoV-2 NN**		--		--

Bold font was used to highlight the different variables analyzed.

**Table 3 vaccines-11-00463-t003:** Fold changes after dose 2 and after booster dose and intragroup comparison.

SARS-CoV-2 Infection Group	Fold after Second Dose (CI 95%)	Fold after Booster Dose (CI 95%)	*p*-Value
SARS-CoV-2 NN	0.35 (0.31–0.39)	0.54 (0.46–0.61)	<0.0001
SARS-CoV-2 NP	0.30 (0.25–0.37)	1.47 (1.15–1.88)	0.057
SARS-CoV-2 PP	0.33 (0.23–0.47)	0.86 (0.62–1.20)	<0.0001
SARS-CoV-2 PN	0.42 (0.31–0.58)	0.54 (0.42–0.71)	<0.0001

## Data Availability

The data presented in this study are available upon request from the corresponding author.

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
