# Peer review of "Impact of BNT162b2 Booster Dose on SARS-CoV-2 Anti-Trimeric Spike Antibody Dynamics in a Large Cohort of Italian Health Care Workers"

_vaccines, 2023, doi:10.3390/vaccines11020463_

Round 1

Reviewer 1 Report

In the paper by Renna et al, the Authors present a serological study of individuals who were vaccinated (and boosted) with the Pfizer mRNA vaccine against Covid-19.  The Authors then analysed antibody titers over the course of the vaccination/period, and presented titer values based on infection history, age and gender.  The Authors ultimately found that individuals who were uninfected prior to vaccination, but were infected during the course of their vaccination regiments had the highest antibody titers compared to all other groups.  The Authors also demonstrated that the change in titer dynamics was largely similar in all vaccination groups.

The Reviewer finds this paper suitable for publication, provided that the Authors address the following issues:

Minor Points: There are a lot of grammatical mistakes in the paper.  Here are only some that were found at first glance:

Line 52- “Firstly approved…” should be changed to “Originally approved…”

Line 73- “…after receiving a booster dose…” or “….after receiving booster doses”

Line 81- “plus the first booster dose”

Line 81- “IgG titers”

Line 92- “…who participated in the vaccine campaign receiving the BNT162b2 mRNA vaccine.”

Line 102-”…day of the administration of the booster dose”

Line 146-“…with a one-dimensional barcode”

Line 398-“After the booster dose…”

Major Points:

Line 133- “Subjects positive for all three anti-N IgG tests without a documented infection after D1 were assumed to have a long-lasting anti-N titer and subsequently assigned to the PN group.”—The Reviewer does not understand why the Authors did this.  Could they please elaborate?  Wouldn’t it be equally possible that the individuals had non-specific reactivity to N-protein (making them effectively NN)?

Figures 4 and 5- Conceptually, the Reviewer has couple of questions about this experiment.  Firstly, why are the figures presented in each age group/gender comprise of all four vaccinated groups put together (i.e. NN, NP, PN and PP)?  Did the Authors analyze the titers each age group/gender by vaccination group?  Were any differences noted?

Author Response

Response to Reviewer 1 Comments

MINOR POINTS:

Line 52- “Firstly approved…” should be changed to “Originally approved…”

Response 1: The sentence has been changed.

Line 73- “…after receiving a booster dose…” or “….after receiving booster doses”

Response 2: The sentence has been changed.

Line 81- “plus the first booster dose”

Response 3: The sentence has been changed.

Line 81- “IgG titers”

Response 4: The sentence has been changed.

Line 92- “…who participated in the vaccine campaign receiving the BNT162b2 mRNA vaccine.”

Response 5: The sentence has been changed.

Line 102-”…day of the administration of the booster dose”

Response 6: The sentence has been changed.

Line 146-“…with a one-dimensional barcode”

Response 7: The sentence has been changed.

Line 398-“After the booster dose…”

Response 8: The sentence has been changed.

MAJOR POINTS:

Point 1: Line 133- “Subjects positive for all three anti-N IgG tests without a documented infection after D1 were assumed to have a long-lasting anti-N titer and subsequently assigned to the PN group.”—The Reviewer does not understand why the Authors did this.  Could they please elaborate?  Wouldn’t it be equally possible that the individuals had non-specific reactivity to N-protein (making them effectively NN)?

Response 1: It should be noted that individuals with all three positive results at Anti-N analysis are only 5. We decided to assign them to the PN group based on 3 factors: I) They have a reported positive swab before D1, thus indicating that the positive anti-N result was not related to a non-specific reactivity to N-protein; II) They did not reported a positive swab after D1 or COVID-19 related symtoms that could indicate a SARS-CoV-2 infection; III) Their anti-Spike IgG titres dynamic was similar to PN individuals.

Point 2: Figures 4 and 5- Conceptually, the Reviewer has couple of questions about this experiment.  Firstly, why are the figures presented in each age group/gender comprise of all four vaccinated groups put together (i.e. NN, NP, PN and PP)?  Did the Authors analyze the titers each age group/gender by vaccination group?  Were any differences noted?

Response 2: We thank the Reviewer for this comment. We decided to present each age or gender group without any subdivision based on infection status since there were no statistical significant differences in gender or in age distribution when comparing the four infection status group (Table 1 and Table 2). Thus we decided to analyse age and gender in univariate model and we had no statistical difference. Likewise for importance of demographic variables as reported in literature, we decide to adjust multivariable model for age and gender.

Reviewer 2 Report

The authors designed this study to investigate the antibody level of different individuals at different age groups after BNT162b2 administration first dose, second dose and booster dose.  Their results show overall that the booster dose elicited a stronger immune response than the first two doses of BNT162b2 and the booster dose slowed down the waning of antibody titer independently of infection status, confirming its importance in preserving a high antibody titer over time. The study design and methods are done properly and the authors written their results clearly and descriptive. I only have few comments for the authors to improve their manuscript before publishing;

1. In abstract line 30 to 32 "Our results suggest that natural infection acquired only after the completion of the vaccination cycle results in higher anti-TrimericS IgG production compared to what occurs in subjects that were infected before and during the vaccination campaign" its not clear and need to rephrase.

2. In Discussion line 413 "Taken together these data" its not clear what the author means.

3. It would be very interesting if the authors have any data or information of antibody level of their subjects and their severity of infection. is it any corelatation  between these two?

Author Response

Response to Reviewer 2 Comments

Point 1: In abstract line 30 to 32 "Our results suggest that natural infection acquired only after the completion of the vaccination cycle results in higher anti-TrimericS IgG production compared to what occurs in subjects that were infected before and during the vaccination campaign" its not clear and need to rephrase.

Response 1: We thank the Reviewer for the suggestion and we have changed the sentence in “Our results suggest that after the booster dose, anti-TrimericS IgG production was higher in subjects infected only after the completion of the vaccination cycle compared to those infected both before and after the vaccination campaign.”

Point 2: In Discussion line 413 "Taken together these data" its not clear what the author means.

Response 2: The sentence has been changed.

Point 3: It would be very interesting if the authors have any data or information of antibody level of their subjects and their severity of infection. is it any corelatation  between these two?

Response 3: We thank the Reviewer for the suggestion. We collected data about COVID-19 infection  based on subjective report of disease symptoms. In subjects infected before vaccination campaign,  symptoms range from mild to severe and some individuals required hospitalization. However, in subjects infected after vaccination, no significative differences in terms of symptoms between individuals have been observed: very mild or mild symptoms and no hospitalization have been reported. For this reason we have not made any stratification based on disease severity.